# Demonstration of ultra-high recyclable energy densities in domain-engineered ferroelectric films

Hongbo Cheng[1,2], Jun Ouyang[1,2,3,4], Yun-Xiang Zhang[1,2], David Ascienzo[3,4], Yao Li[1,2], Yu-Yao Zhao[1,2] & Yuhang Ren[3,4]

Dielectric capacitors have the highest charge/discharge speed among all electrical energy devices, but lag behind in energy density. Here we report dielectric ultracapacitors based on ferroelectric films of $Ba(Zr_{0.2},Ti_{0.8})O_3$ which display high-energy densities (up to $166\,J\,cm^{-3}$) and efficiencies (up to 96%). Different from a typical ferroelectric whose electric polarization is easily saturated, these $Ba(Zr_{0.2},Ti_{0.8})O_3$ films display a much delayed saturation of the electric polarization, which increases continuously from nearly zero at remnant in a multipolar state, to a large value under the maximum electric field, leading to drastically improved recyclable energy densities. This is achieved by the creation of an adaptive nano-domain structure in these perovskite films via phase engineering and strain tuning. The lead-free Ba $(Zr_{0.2},Ti_{0.8})O_3$ films also show excellent dielectric and energy storage performance over a broad frequency and temperature range. These findings may enable broader applications of dielectric capacitors in energy storage, conditioning, and conversion.

[1] Key Laboratory for Liquid–Solid Structural Evolution and Processing of Materials (Ministry of Education), School of Materials Science and Engineering, Shandong University, Jinan 250061, China. [2] Suzhou Institute of Shandong University, Suzhou 215123, China. [3] Physics and Astronomy, Hunter College of the City University of New York, 695 Park Avenue, New York, NY 10065, USA. [4] The Graduate Center, The City University of New York, 365 5th Avenue, New York, NY 10016, USA. Yun-Xiang Zhang and David Ascienzo contributed equally to this work. Correspondence and requests for materials should be addressed to J.O. (email: ouyangjun@sdu.edu.cn) or to Y.R. (email: yre@hunter.cuny.edu)

Dielectric capacitors, as compared with batteries and other devices for electrical energy storage, excel in specific power, compactness, and cost-effectiveness[1–3]. These features have created a broad spectrum of applications for them in microelectronics and electric power systems[4–7]. Recently, due to the high electrical energy densities originated from their large dielectric constants (Supplementary Note 1), ferroelectric dielectric films have attracted intensive research interest[1,2,8–12]. A typical ferroelectric has a large remnant polarization ($P_r$) and a slightly larger saturated polarization ($P_s$), leading to the formation of a square-shaped polarization (**P**)-electric field (**E**) hysteresis loop. In addition, the saturation of its polarization usually occurs at a field well below its electrical breakdown field. Consequently, it shows an underachieved energy density ($W_C$) with an increasing electric field, and a much poorer energy efficiency ($\eta$), when compared to a linear dielectric (Supplementary Note 1).

A high-energy density and efficiency (i.e., high recyclable energy density $W_C \cdot \eta$) can be simultaneously achieved in a ferroelectric film with a slim **P**–**E** hysteresis loop, which features a small $P_r$ and a large $P_s$. Furthermore, if we can substantially delay its polarization saturation[1], the ferroelectric film can store significantly more electrical energy. Several approaches to slim down the **P**–**E** loop of a ferroelectric film have been reported, including using relaxor ferroelectrics[1,8] or compositions near phase boundaries[10], utilizing space-charges[13] or interfaces[14], and the inclusion of a dead layer[15]. Using these methods, $W_C$ of ferroelectric films has been rapidly increased to a level of 20–70 J cm$^{-3}$[2,8,14,15], an order of magnitude higher than their bulk counterparts. The energy efficiency has also been improved to the level of 60–80%[8,14,15].

In this work, a rhombohedral Ba(Zr$_{0.2}$,Ti$_{0.8}$)O$_3$ (BZT) ceramic[16] target was used to deposit epitaxial films with different thicknesses (350–1.8 μm) on a variety of (100) cubic/pseudocubic single crystalline substrates, including LaAlO$_3$ (LAO), (La,Sr)(Al,Ta)O$_3$ (LSAT), and SrTiO$_3$ (STO) ($a_S$ ~3.79–3.905 Å)[17]. On these substrates, a BZT film will have a metastable tetragonal phase. Furthermore, the large lattice constant of BZT (~4.06 Å) allows us to engineer an in-plane compressive misfit decaying along the thickness direction of the film, thus enabling the creation of competing polydomain structures for better energy storage performance (Supplementary Note 2). The demonstrated record-high capacitive energy density $W_C$ (~166 J cm$^{-3}$), and drastically improved charge–discharge efficiency $\eta$ (up to ~96%), together with a low dielectric loss and a high dielectric stability, indicate that the utilization of an optimal polydomain structure could become a general strategy to enhance the capacitive performance of ferroelectrics.

## Results

**Design of a microstructure.** Here, we describe a microscopic approach to further improve both energy density and efficiency of a ferroelectric film, through the design of self-assembled, energy-absorbing polydomain nanostructures[18,19]. In a single-domain film, the unipolar state at saturation (absent of 180° domains) is readily achieved under an external electric field, which can only be partially depoled by discharging, leading to a poor recyclable energy density (Fig. 1a, b). On the other hand, an engineered polydomain film with an inclined "head-to-tail" polarization configuration in the neighboring domains with respect to the direction of **E**, demands a much larger electric field to be fully poled, resulting in a much higher $W_C$. Furthermore, an outstanding energy efficiency $\eta$ will also be achieved due to a zero net polarization at the remnant state, i.e., $P_r$ ~ 0 (Fig. 1c).

The most common polydomain structures in ferroelectric films are the "$a/c$ polytwins" or "90° domains" observed in (001)

tetragonal ($T$) films[19,20]. It is difficult to transform the $a/c$ domains into one another reversibly in constrained films due to a large spontaneous strain (a spontaneous strain, or a self-strain, is a lattice distortion of a crystalline solid due to a phase transformation or a transformation between twin variants—it characterizes the relative differences of the lattice parameters between the product phase/domain and the parent phase/domain), as well as an interlocking three-domain architecture[20]. Furthermore, the remnant polarization of an $a/c$ polytwin is of a finite value due to the $c$ domain contribution. In contrast, a

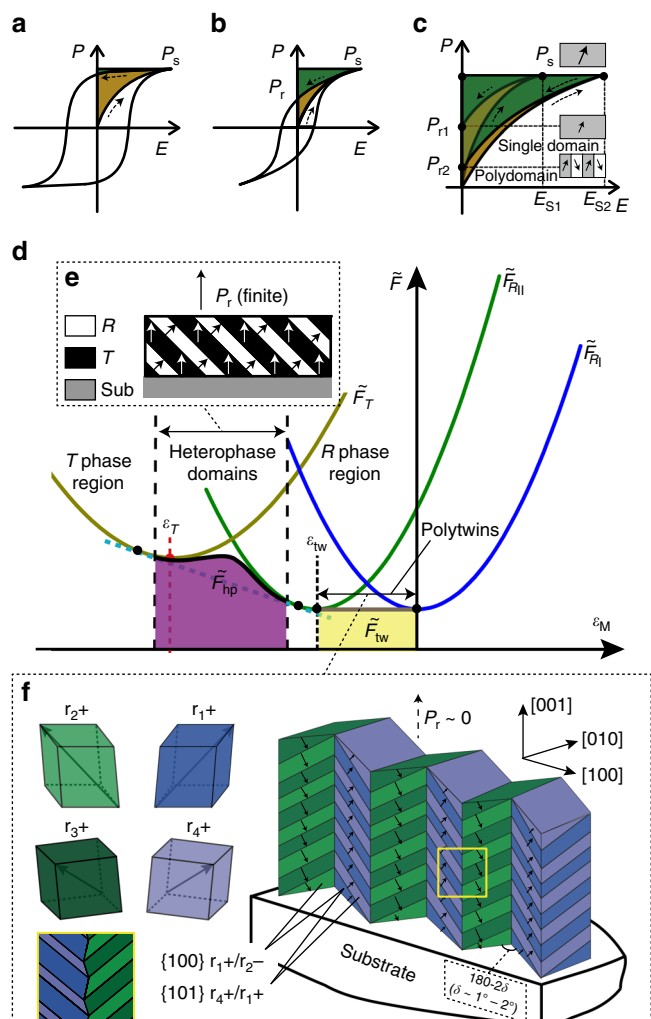

**Fig. 1** Energy storage in ferroelectrics and optimal design of their domain structures. **a** A normal ferroelectric **P**–**E** loop, **b** a slim ferroelectric **P**–**E** loop, and **c** **P**–**E** loops (partially shown) for a single-domain and an engineered polydomain ferroelectric film. The green-shaded areas represent the recyclable energy densities, while the shaded areas within the loops are the energy losses of one charge–discharge cycle. **d** Free energies ($\tilde{F}$) vs. misfit strain $\varepsilon_M$ for a film with a single-domain rhombohedral phase ($R_I$, $\tilde{F}_{R_I}$ or $R_{II}$, $\tilde{F}_{R_{II}}$), a single-domain tetragonal phase ($T$, $\tilde{F}_T$), $T/R$ heterophase polydomains ($\tilde{F}_{hp}$), and rhombohedral polytwins ($R_I/R_{II}$, $\tilde{F}_{tw}$). Stable regions of the polydomain structures are determined by the common tangent method applied to the free energy curves of single-domain films. $\varepsilon_T$ and $\varepsilon_{tw}$ are the spontaneous strains of the $T$ and $R_{II}$ domains with respect to the $R_I$ domain. Schematics of the polydomain structures are shown in **e** for a $T/R$ heterophase polydomain, and **f** for hierarchical rhombohedral polytwins formed in a (001)-oriented epitaxial ferroelectric film with a bulk rhombohedral phase

rhombohedral (*R*) ferroelectric has a polytwin structure with small spontaneous strains between the domain variants. In a (001) epitaxial film, the *R* polytwins with {100}-type domain boundaries display the desirable polarization configuration, as shown in Fig. 1c[21]. Under charging/discharging electric fields, films with such a domain structure can reversibly store and release a large amount of energy, by transforming back and forth between the remnant, charge-free polydomain state and a poled,

highly charged state in the process towards saturation (Fig. 1c). Another type of polydomain structures is the so-called "hetero-phase domains", i.e., polydomain structures involving a strain-stabilized component phase, which usually form in thin films to accommodate their large misfit strains[22,23]. By transforming into a single-phase structure under an electric field[22], the heterophase domains absorb additional electrical energy and thus delay saturation of the overall polarization.

In Fig. 1d, free energies of two single-domain states of a rhombohedral phase ($R_I$, $R_{II}$), a single-domain tetragonal phase (*T*), as well as a $R_I/R_{II}$ polytwin structure and a *T*/*R* heterophase polydomain structure, are presented as functions of the biaxial misfit strain $\varepsilon_M$ (using $R_I$ as the reference state) of an epitaxial ferroelectric film. The two types of polydomain structures have different stability regions and energy storage characteristics. For a given initial misfit (substrate), thin films with dominant heterophase polydomains (Fig. 1e) will show higher energy densities due to a large applicable electric field, while thick films with dominant $R_I/R_{II}$ polytwins will show improved energy efficiencies due to a diminishing $P_r$. Moreover, it is possible to form hierarchical polytwins in thick films due to the trend of total strain relaxation[24,25]. For a (001) rhombohedral film with four possible domain variants, such a structure is constructed by the alternation of two polytwins growing along the film-thickness direction with {100} domain boundaries, each consisting of two domain variants assembled at the nanoscale with {101} domain boundaries (Fig. 1f).

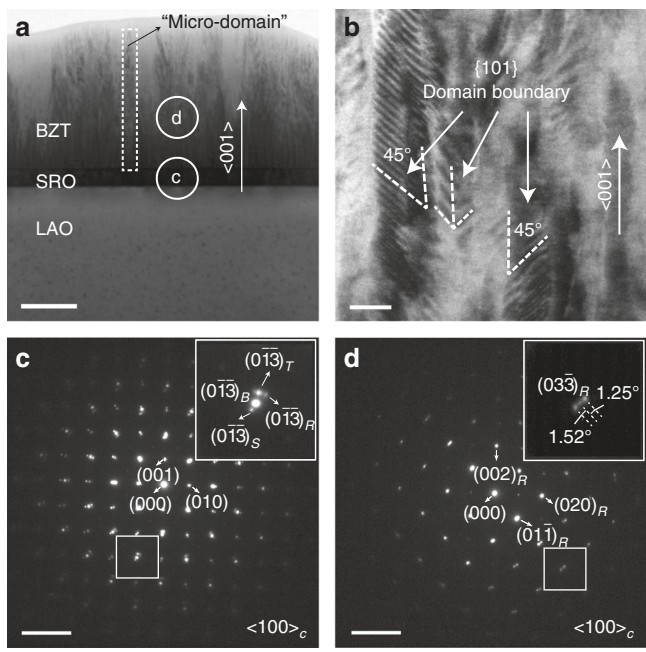

**Fig. 2** Transmission electron microscopy (TEM) analysis. **a**, **b** Cross-sectional TEM images of a 0.7-μm-thick BZT film (scale bar: 200 nm for **a** and 20 nm for **b**), and **c**, **d** SAED patterns (scale bar: 5 nm⁻¹) for the areas marked in **a**. Subscripts *T* and *R* denote the *T* and *R* phases of the film, while *B* and *S* represent the bottom electrode and the substrate

**Structural analysis.** The (*l*00) epitaxial growth and a mixed *T*/*R* phase structure were revealed in all BZT films via X-ray diffraction analysis (Supplementary Note 3, Supplementary Fig. 3) and transmission electron microscopy (TEM) (Fig. 2, Supplementary Note 4, Supplementary Fig. 4). Figure 2a, b show bright-field cross-sectional TEM images of a 0.7-μm-thick BZT film. A polydomain microstructure with {100} domain boundaries is visualized in Fig. 2a with the domains grown up-straight from the bottom interface to the film surface. Furthermore, in Fig. 2b, second-order "nano-domains" with {101} boundaries are observed

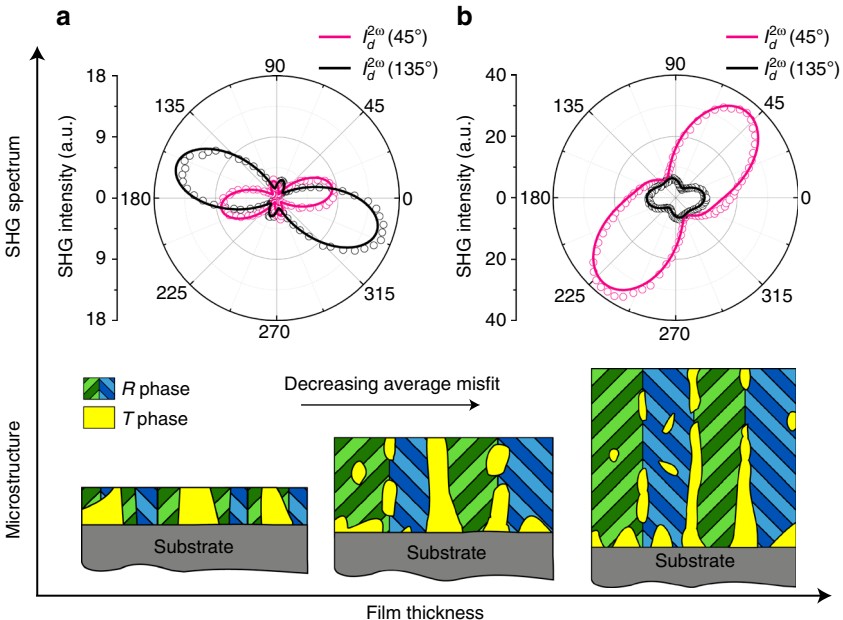

**Fig. 3** Schematic of the microstructure evolution with film thickness. **a**, **b** are intensity profiles of the transmitted *d*-polarized SHG signals from a 350-nm and a 1.8-μm-thick BZT film, respectively

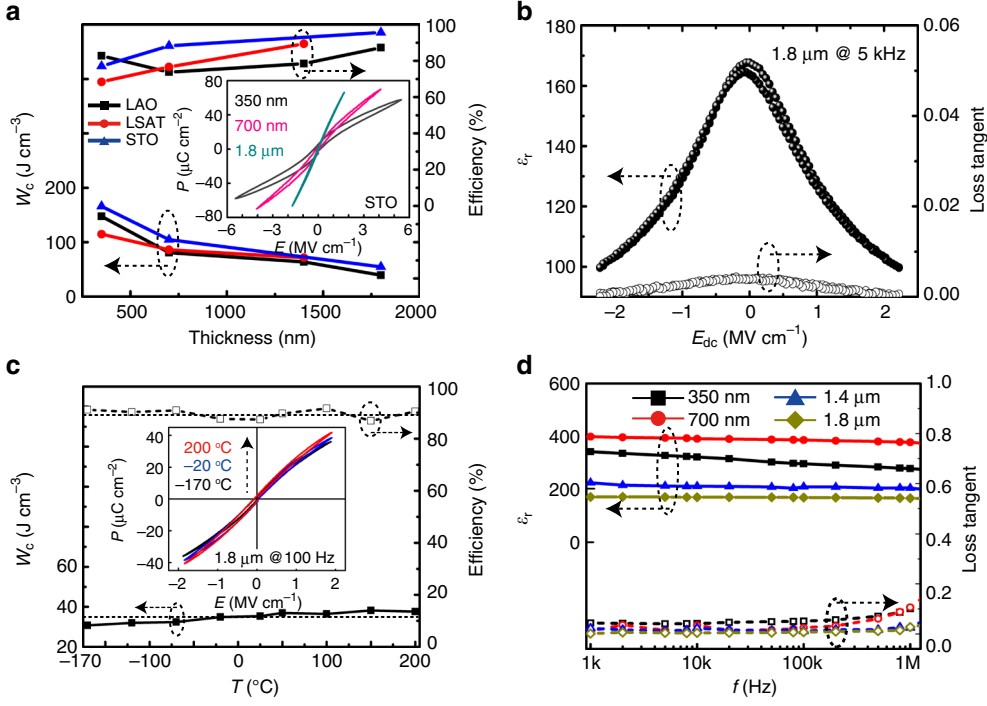

**Fig. 4** Electrical energy storage capability and dielectric stability. **a** Thickness-dependent energy storage densities and efficiencies ($W_C$ and $\eta$) of the BZT films grown on different substrates, the inset shows the typical **P**–**E** loops. **b** Room temperature $\varepsilon_r$-**E** and $tg\delta$-**E** curves of a 1.8-μm-thick BZT film (on LAO). **c** $W_C$ and $\eta$ as functions of temperature for the film in **b**, computed from its temperature-dependent **P**–**E** loops (inset). **d** Room temperature $\varepsilon_r$–$f$ and $tg\delta$–$f$ curves for BZT films grown on the same substrates (LAO) but with different thicknesses

inside the columnar "micro-domains" shown in Fig. 2a. These observations indicate the formation of a hierarchical polytwin structure. Figure 2c, d show the selected area electron diffraction (SAED) patterns. While Fig. 2c confirms the epitaxial growth and a *T*/*R* heterophase microstructure near the bottom interface of the BZT film, Fig. 2d shows a *R* phase with split diffraction spots, verifying a dominant microstructure of rhombohedral polytwins in the film bulk. It is noted that the tetragonal domains, while abundant near the bottom interface of the films, are only scarcely observed in the film bulk (Supplementary Fig. 4). This thickness-dependent microstructure is attributed to the formation of heterophase polydomains and polytwins (homophase polydomains) as competing mechanisms to relax the misfit stresses between the BZT film and the substrate. While the heterophase polydomain structure dominates in the case of large misfit strains/thin films, the hierarchical polytwin structure takes control after the *T* phase has gradually transformed into the bulk *R* phase due to stress relaxation with an increasing film thickness[26]. In Fig. 3, sketches of the thickness-dependent polydomain structures are presented, together with the characteristic spectra of optical second-harmonic generation (SHG) polarimetry. The transmitted *d*-polarized SHG signal intensity increases with the film thickness, reflecting an increasing amount of the rhombohedral domains with a decreasing average misfit[27,28]. The SHG results (Supplementary Note 5, Supplementary Fig. 5) are in good agreement with the TEM observations, both supporting the proposed microstructure model of "competing polydomains".

**Energy storage and dielectric properties and stability**. Figure 4a summarizes energy storage performance ($W_C$ and $\eta$) of the BZT films at room temperature (Supplementary Table 1, Supplementary Figs. 1, 2). Films on all three substrates display excellent

energy storage characteristics: the 350-nm films show very high-energy densities (~115–166 J cm⁻³) due to a high applicable electric field (~5.7 MV cm⁻¹, inset of Fig. 4a), as well as good energy efficiencies (~69–83%); while the 1.8-μm films demonstrate record-breaking energy efficiencies for ferro-electrics (~88–96%), together with a well-scaled high-energy density (~40–55 J cm⁻³). Among the three different substrates, STO has the smallest initial misfit with bulk BZT(~3.8%). Favorable lattice matching between the two ensures that the thickness-dependent polydomain structure is the dominant mechanism of stress relaxation in the film. Better energy storage capabilities are expected in BZT films grown on STO, which was supported by the characterization results (Fig. 4a). It is noted that the tuning effect of the substrate on energy storage is more prominent in thinner films, which have higher residual strains.

Moreover, the inset of Fig. 4a displays the representative **P**–**E** loops of the BZT films (grown on the same substrates but with different thicknesses). Slim loops with high and unsaturated $P_{max}$ (~58–70 μC cm⁻²), and low $P_r$ (~1.5–5.5 μC cm⁻²) values are consistently shown by all films, confirming the positive role played by the polydomain structures on reducing the **P**–**E** hysteresis and delaying the polarization saturation (Fig. 1c). Moreover, the thickest film displays a diminishing remnant polarization (~1.5 μC cm⁻²), resulting in the slimmest **P**–**E** loop and highest $\eta$ (~96%) among the BZT films. All these observations support our design of the polydomain structure for better energy performance. Furthermore, the unsaturated **P**–**E** loops indicate future opportunities to further improve the energy storage capabilities of ferroelectrics via the polydomain approach.

In Fig. 4b, the room temperature relative dielectric constant $\varepsilon_r$ and loss tangent $tg\delta$ are presented as functions of the dc bias field $E_{dc}$. The low dielectric losses ($tg\delta$ < ~0.5%) of the BZT film,

measured in a broad range of $E_{dc}$ (0 ~2.2 MV cm$^{-1}$, or 400 V on a 1.8-μm film), are consistent with its superior energy efficiency. Moreover, **P**–**E** loops of this BZT film (at $E_{max} = 1.9$ MV cm$^{-1}$) were measured at temperatures between −170 and 200 °C, and the results are presented in Fig. 4c. Energy density $W_C$ and efficiency $\eta$ only show small variations in this broad temperature range (~±10% for $W_C$ and ±3% for $\eta$). While the stable capacitive performances at low temperatures can be attributed to the close free energies of the $T$ and $R$ phases, those at high temperatures are due to an elevated Curie temperature (>450 °C, Supplementary Note 6, Supplementary Fig. 6), which can be attributed to the effect of a large misfit strain[29,30].

Finally, Fig. 4d presents the frequency dispersion behavior of the dielectric property of the BZT films. With an increasing thickness, the BZT films showed reduced frequency dependences in $\varepsilon_r$ and $tg\delta$, as well as a reduced loss value. The thickest film (1.8 μm) shows an almost frequency-independent $\varepsilon_r$ and a small loss (≤1.5%) in the measuring frequency range of (1 kHz, 1 MHz) (Supplementary Fig. 7). This thickness-dependent dielectric behavior is attributed to the evolution of polydomain structure with film thickness (Fig. 3). The hierarchical $R$-phase polydomains, which are dominant in thicker films, can easily adapt to an $ac$ electric field by adjusting the variant ratios of the nanotwins. Furthermore, a fatigue test with $10^8$ cycling times showed ~80% retention of the energy storage capability, as well as an improved energy efficiency of the BZT thin-film capacitors (Supplementary Note 7, Supplementary Fig. 8). This can be attributed to the transformation of the metastable $T$ phase into the stable polydomain $R$ phase under a cycling E field. Overall, the desirable combination of frequency-insensitive capacitive performance, high recyclable energy density, and good temperature stability in such thin-film dielectrics will transform the way dielectric capacitors are used for energy applications.

## Discussion

The long-range stress field in an epitaxial ferroelectric film can be relaxed by the formation of polytwins or heterophase polydomains[19–26]. These domains are "elastic domains"[18] by the nature of their origin. In the absence of an external field, the formation of a polydomain structure is driven by a reduction in misfit elastic energy due to the indirect domain elastic interaction through the substrate[31], which manifests itself by an energy term

$$e^I(\vec{P}_1, \vec{P}_2) = \frac{1}{2}\hat{\varepsilon}_{12} \cdot G(\vec{n}) \cdot \hat{\varepsilon}_{12} \qquad (1)$$

Here, superscript "$I$" denotes "indirect domain elastic interaction", $\hat{\varepsilon}_{12}$ is the spontaneous strain tensor between domain variants 1 and 2 dependent on their polarizations $\vec{P}_1$ and $\vec{P}_2$, and $G(\vec{n})$ is the planar elastic modulus tensor dependent on the film normal $\vec{n}$[31]. Since we consider a fixed $\vec{n} = [001]$ in the present discussion, the reduced misfit energy, $-\alpha(1-\alpha)e^I(\vec{P}_1, \vec{P}_2)$, is a function of the domain fraction $\alpha$ (of variant 2) and the polarizations. The equilibrium domain fraction of a polytwin structure, $\alpha_0$, can be estimated by[31]

$$\alpha_0 = \frac{1}{2}\left[1 - \frac{\Delta\tilde{F}}{e^I}\right], \qquad (2)$$

where $\Delta\tilde{F} = \Delta e_0 = e_2(\vec{P}_2, \hat{\varepsilon}_M) - e_1(\vec{P}_1, \hat{\varepsilon}_M)$ is the difference in free energy density due to the change in misfit elastic energy density by transforming a single-domain structure of variant 1 ($\vec{P}_1, e_1$) into that of domain variant 2 ($\vec{P}_2, e_2$). Here, $e_1$, $e_2$ are the misfit elastic energy densities of single-domain structures of

domain variant 1 and 2, respectively, which are functions of their polarizations and a misfit of the film with the substrate, $\hat{\varepsilon}_M$ ($\hat{\varepsilon}_M$ is a strain tensor and $\varepsilon_M$ is its nonzero, symmetrical diagonal component).

Based on Eq. (2), we have $|\Delta\tilde{F}| = |\Delta e_0| < e_0^I$ for a stable poly-twin structure ($0 < \alpha_0 < 1$). This implies that, if the difference in misfit elastic energy density between the domain variants, $\Delta e_0$, is smaller than the energy of indirect domain elastic interaction, $e^I$, then, the polytwin structure is more stable than a single-domain one. This corresponds to a misfit range of $(0, \varepsilon_{tw})$, where $\varepsilon_{tw}$ is the in-plane spontaneous strain between the twin variants (subscript "tw" denotes "twinning"). In Fig. 1d, the stable region for poly-twins is schematically shown as the yellow-shaded area in the free energy ($\tilde{F}$)–misfit strain ($\varepsilon_M$) diagram. For a rhombohedral fer-roelectric like BZT, the twinning spontaneous strain is small (≤1%, which is even smaller in higher-order, hierarchical poly-twin structures[24]), and hence the polytwin structure is expected to be abundant in thick films or relaxed regions of a thin film (Figs. 2 and 3, Supplementary Fig. 4).

Formation of heterophase polydomain structure (HPS) in a ferroelectric film is much less observed in experimental studies due to the extra energy barrier associated with the incompatibility between the phases[23], $\alpha(1-\alpha)e^{DI}(\vec{P}_1, \vec{P}_2)$ (superscript "DI" denotes "direct domain interaction"), where

$$e^{DI}(\vec{P}_1, \vec{P}_2) = e_{12}(\vec{P}_1, \vec{P}_2) + [(\vec{P}_2 - \vec{P}_1) \cdot \vec{m}]^2/(2\varepsilon_0\varepsilon_b), \qquad (3)$$

consisting of elastic energy $e_{12}(\vec{P}_1, \vec{P}_2)$ due to strain incompat-ibility between the domains and electrostatic energy $[(\vec{P}_2 - \vec{P}_1) \cdot \vec{m}]^2/(2\varepsilon_0\varepsilon_b)$ due to uncompensated polarization charges across their interface $\vec{m}$ ($\varepsilon_b$ is the background dielectric constant and $\varepsilon_0$ is the vacuum dielectric constant). Taking into account bulk free energy difference and the energy due to domain incompatibility, the equilibrium domain fraction $\alpha_0$ for a HPS is[23]

$$\alpha_0 = \frac{1}{2}\left[1 - \frac{\Delta\tilde{F}}{e^I - e^{DI}}\right] \qquad (4)$$

where $\Delta\tilde{F} = \Delta f_0 + \Delta e_0$, $\Delta f_0 = f_2(\vec{P}_2) - f_1(\vec{P}_1)$ and $\Delta e_0$ are dif-ferences in bulk free energy and misfit elastic energy densities between domain variants 1 and 2, respectively. To form a stable HPS in the as-grown film, $e^{DI} < e^I$ and $-1 < \frac{\Delta\tilde{F}}{e^I - e^{DI}} < 1$, implying a small $\Delta f_0$. In a free energy ($\tilde{F}$)–misfit strain ($\varepsilon_M$) diagram, the stable region of a HPS lies between those of the strain-stabilized phase (metastable phase) and the bulk phase, corresponding to a misfit range roughly between the two spontaneous strains. Since the spontaneous strain of the metastable phase is usually larger than the twinning spontaneous strain of the bulk phase, hetero-phase polydomain structures are usually formed in thin films or highly stressed regions of a thick film (Figs. 2 and 3, Supple-mentary Fig. 4). In Fig. 1d, the stable region for a $T/R$ heterophase polydomain structure is schematically shown as the purple-shaded area near [$\varepsilon_T$, $\varepsilon_{tw}$] in the $\tilde{F}$ vs. $\varepsilon_M$ diagram.

So far, only in BiFeO$_3$[22] and PZT films[32], ferroelectric het-erophase polydomain structures have been experimentally observed and reported. In BaTiO$_3$, there is a rich spectrum of polymorphic phases with the transformation temperatures well below room temperature, i.e., their bulk free energies are widely apart near room temperature. By doping with larger B-site ions of Zr$^{4+}$, free energies of the rhombohedral and the tetragonal phases are drawn closer to each other (Supplementary Note 8, Supple-mentary Fig. 9), allowing further manipulation via misfit strains to form a heterophase polydomain structure (Fig. 1d, Supple-mentary Note 9). Furthermore, the enlarged lattice parameter of BZT (~4.06 Å), as compared to BaTiO$_3$ (~3.99 Å), promotes the

formation of HPS in films with a wide range of thickness on the commonly used perovskite substrates ($a_S$ ~3.7–3.9 Å). These have allowed us to create a thickness-dependent microstructure consisting of competing polydomain nanostructures (polytwins and HPS, as shown in Fig. 3) for a better energy storage performance.

## Methods

**Materials**. Single-crystalline substrates of LaAlO$_3$, (La,Sr)(Al,Ta)O$_3$ and SrTiO$_3$ (10 mm × 10 mm × 0.5 mm), as well as the SrRuO$_3$ ceramic target ($\Phi = 50$ mm, $t = 5$ mm, 3$N$ purity) were provided by Anhui Institute of Optics and Fine Mechanics, Chinese Academy of Sciences, China. The Ba(Zr$_{0.2}$Ti$_{0.8}$)O$_3$ (BZT) ceramic target was prepared in-house with the same shape and size as the SrRuO$_3$ target via a solid-state reaction method. A single-phase rhombohedral structure was identified in the BZT ceramic with a lattice parameter of ~4.06 Å via X-ray diffraction (XRD) analysis.

**BZT film growth**. RF-magnetron sputtering was used for the growth of the epitaxial polydomain films. A base pressure of $2.0 \times 10^{-4}$ Pa was achieved in a multitarget sputtering chamber prior to the sequential deposition of a bottom electrode (SrRuO$_3$) layer and a ferroelectric film. BZT films in the thickness range of 350–1800 nm and a SrRuO$_3$ layer of ~100-nm thick were sputtered from their corresponding ceramic targets. During the sputtering process, the deposition temperature was held at 650 °C, while the chamber pressure was kept at 1.4 Pa in a mixed Ar/O$_2$ atmosphere (Ar/O$_2$ flow ratio = 4:1). Metal–ferroelectric–metal (MFM) testing structures were formed after deposition of top electrodes of circular Au pads ($\Phi = 200$ μm) (sputtered at room temperature via a shadow mask).

**Characterization**. The phase structures and crystallographic orientations of the BZT films were analyzed via XRD using a commercial Rigaku Dmax-rc diffractometer (regular $\theta$–$2\theta$ scans), and a high-power IP crystal X-ray diffractometer equipped with R-Axis Spider ($\Phi$-scans). Phase structures and nanostructures of the BZT films were also investigated via transmission electron microscopy (TEM) using a JEM-2010 microscope (JEOL, Tokyo, Japan). The pseudostatic ferroelectric hysteresis loops (P–E loops from the polarization-voltage tests) of the BZT films were measured by using a RT-Precision LC ferroelectric testing system (Radiant Technology, NM, USA; $f = 1$ kHz for the 350-nm films, and $f = 100$ Hz for thicker films), the dielectric properties ($\varepsilon_r$–$f$/tg$\delta$–$f$ from capacitance–frequency tests, i.e., C–$f$ tests; $\varepsilon_r$-**E**/tg$\delta$-**E** from capacitance–voltage tests, i.e., C–V tests) were measured by using a high-precision digital bridge (LCR meter, QuadTech 7600plus), and an Agilent1505A power device analyzer. The C–V tests were carried out by superimposing a 50-mV, 5-kHz AC signal on a DC bias voltage sweeping from its negative maximum to its positive maximum, and vice versa. The temperature-dependent P–E loops and $\varepsilon_r$-T/tg$\delta$–T characteristics (from the capacitance–temperature tests, i.e., C–T tests) were measured by using a temperature-controlled probe station (Linkam-HFS600E-PB2). SHG measurements were carried out at room temperature using a mode-locked Ti:sapphire ultrashort pulse laser (80 MHz, 10 nJ per pulse, 100 fs) and a regenerative amplifier (250 kHz) as the fundamental light source ($\lambda = 810$ nm) with a beam spot of ~10 μm in diameter for reflected and transmitted signal detections, respectively.

**Data availability**. All relevant data are available from the first author and corresponding authors upon request.

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

## Acknowledgements

We acknowledge the financial support of NSFC (Grant Nos. 51772175, 91122024), and AFOSR (Grant Nos. FA9550-17-1-0342, FA9550-17-1-0339). J.O. acknowledges the support from the Program for New Century. Excellent Talents in University (Ministry of Education), the Nanotechnology Projects of Soochow City (Grant No. ZXG201445), and the Fundamental Research Funds of Shandong University (Grant Nos. 2017ZD008, 2015YQ009 and 2015JC034). We thank Professor Manfred Wuttig (University of Maryland, College Park) for many fruitful discussions.

## Author contributions

J.O., Y.R., and H.C. devised the original concept and cowrote the manuscript. H.C. prepared the ceramic target and films, performed room temperature electrical measurements, collected XRD data, and analyzed the TEM data. Y.-X.Z. helped to prepare the TEM samples and performed the temperature-dependent electrical measurements. Y.L. and Y.-Y.Z. were

responsible for high-field electrical measurements. D.A. and Y.R. were responsible for the SHG data, and D.A. also proofread and commented on the manuscript.

## Additional information

**Competing interests:** The authors declare no competing financial interests.

