## [Peer Review File · Nature Communications]

Reviewer #1 (Remarks to the Author):

I commented before that after the two reviews I would find it acceptable for Nature communications.

Certainly this new version is again much improved. The reader should be able to understand the novel concept. The properties are well presented and put well into context. The structural characterization supports the concept very well.

I have almost no further comments/request, except for something very minor:

I. ref. 14 and 15 in the manuscript do not have a page or manuscript number.

II. In suppl. on page 12, line 203 it reads: predominately

III: In Fig. S.9 the frequencies are given in kHz instead of in kHz.

Reviewer #2 (Remarks to the Author):

The authors describe a new way to achieve high energy densities in the ferroelectric phase of a solid state capacitor. By choosing a substrate with a suitable misfit strain with respect to the film, the authors obtained films with a suppressed saturation of polarization and suppressed remanent polarization. This resulted in a large energy storage efficiency and energy density of the capacitors. The authors have performed new experiments to address my concerns expressed in an earlier review.

I find the idea interesting and results novel.

I am distracted by the expression "self-strain" used instead of usual term "spontaneous strain" (line 262). I do not understand why the authors wish to coin a new term, which I find confusing, when a perfectly good expression already exists.

REVIEWERS' COMMENTS:

We thank the reviewers for their comments and suggestions of revisions which have helped improve the quality of our manuscript. In the following, we respond to the comments of the reviewers in their original order.

Reviewer #1 (Remarks to the Author):

I commented before that after the two reviews I would find it acceptable for Nature communications.

Certainly this new version is again much improved. The reader should be able to understand the novel concept. The properties are well presented and put well into context. The structural characterization supports the concept very well.

I have almost no further comments/request, except for something very minor:

i. ref. 14 and 15 in the manuscript do not have a page or manuscript number.

We thank the reviewer for pointing this out for us. We have made all related corrections. Please see the revised references 14 and 15, respectively.

ii. In suppl. on page 12, line 203 it reads: predominately

Yes, we thank the reviewer for pointing this out. We have changed the word into “predominantly”.

iii: In Fig. S.9 the frequencies are given in kHz instead of in kHz.

Sorry for the mistake, we have corrected it.

Reviewer #2 (Remarks to the Author):

The authors describe a new way to achieve high energy densities in the ferroelectric phase of a solid state capacitor. By choosing a substrate with a suitable misfit strain with respect to the film, the authors obtained films with a suppressed saturation of polarization and suppressed remanent polarization. This resulted in a large energy storage efficiency and energy density of the capacitors. The authors have performed new experiments to address my concerns expressed in an earlier review.

I find the idea interesting and results novel.

I am distracted by the expression "self-strain" used instead of usual term "spontaneous strain" (line 262). I do not understand why the authors wish to coin a new term, which I find confusing, when a perfectly good expression already exists.

We thank the reviewer for pointing this out. Per his/her input, we have made the corrections, i.e., replacing the less used term "self-strain" with "spontaneous strain" throughout the manuscript and the SOM document.